# An Integration Model for Flux Density Distribution Formed by a Heliostat

**Chenggang Zong [1], Yemao Shi [2], Liang Yu [2], Bowen Liu [3] and Weidong Huang [3,*]**

1 Asset Management Co., University of Science and Technology of China, 96 Jinzhai Road, Hefei 230026, China
2 School of Earth and Space Science, University of Science and Technology of China, 96 Jinzhai Road, Hefei 230026, China
3 Department of Environmental Science and Engineering, University of Science and Technology of China, 96 Jinzhai Road, Hefei 230026, China
* Correspondence: huangwd@ustc.edu.cn; Tel.: +86-551-63606631; Fax: +86-551-63607386

**Abstract:** An accurate flux density calculation is essential for optimizing and designing solar tower systems. Most of the existing methods introduce multiple assumptions, and the accuracy and scope of the application are limited. This paper proposes an integration model used to calculate the flux density distribution after only applying the Gaussian model for solar brightness distribution. It is the first time that multiple reflections and the influence of the optical error transferred from different planes of the glass mirror are considered in order to build an optical model for the flux density of a heliostat. The reflection from two surfaces of the glass mirror used to form three main parts of beams was considered in the present model, and Fresnel's equations were applied to calculate the energy of the three parts of reflected rays. An elliptic Gaussian model was applied for the optical error distribution of the heliostat. The model error was evaluated using the experimental data of ten heliostats, and the applicability and accuracy of the model were verified through flux distribution and an intercept factor. The average relative prediction error of the present model from the experimental data was only 2.83%, which is less than SolTrace and other models.

**Keywords:** Gaussian distribution; direct integration method; Fresnel's equations; solar flux density; optical error





## 1. Introduction

The accurate prediction of a solar flux density distribution formed by a heliostat has attracted widespread attention. It can avoid an excessive solar energy density and calculate the receiver's intercept factor and optical efficiency to the focusing solar spot, which can be used to design and optimize the heliostat field [1,2].

The simplest calculation method uses functions to describe the flux density distribution on the receiver plane directly. The UNIZAR function proposed by Collado et al. [3] and the HFCAL function proposed by Michael Kiera [4] is simple and fast in its calculation. Still, there is a significant error in calculation results compared with the experimental data.

Current methods for calculating the heliostat flux density mainly include the convolution integration method and ray-tracing method [5]. The ray-tracing method is based on the law of light reflection used to track the path of incident rays in the system. The Gaussian model is mainly used to describe the optical error of a heliostat in order to bring errors. However, it still requires extensive calculation and is not suitable for optimization. The convolution integration method is an indirect integration method that regards the flux density on the image plane as the convolution of the solar brightness distribution and image function of the heliostat. It then projects the flux density distribution on the image plane onto the receiver plane. The convolution integration is the approximation of the actual physical image. At the same time, the projection process also introduces more errors as it assumes that the solar ray is entirely from the center of the heliostat. Although the

calculation speed is faster than the ray-tracing method, it is sometimes not accurate enough to calculate the flux density distribution of a single heliostat. However, it is more accurate than assuming that all incident radiation has been intercepted by the receiver [6].

Lipps [7] comprehensively considered the solar brightness distribution and the elliptic Gaussian optical error distribution and solved the flux density distribution by using the convolution method. The HELIOS [8] system uses two-dimensional Fourier transform to calculate the error of the reflected ray generated by the ellipse Gaussian distribution of the surface error in the face of different solar intensity distribution functions. The optical error distribution function of an ellipse Gaussian has been proven to be more consistent with the actual data and applicable to a wider range of situations to a certain extent. However, the above methods do not consider the transmission of the optical error from the heliostat to reflected light, which is different from the actual situation. Previous studies have shown that even if the optical error is the same at different parts of the heliostat, the optical error transferred to the reflected ray is different [9], resulting in a difference from the actual situation.

Before, we applied the convolution method to deduce an equation used to simulate the solar flux of a heliostat [10]. Its calculation speed is fast, and its calculation data are relatively accurate. However, it introduces the Gaussian function for a solar brightness distribution and an image of the heliostat and convolution methods, and the scope of the application is limited.

The convolution method assumes that the image of the heliostat is similar to its shape. Suppose that the ratio of the heliostat size to the distance from the receiver or the incident angle is too large. In this case, it will bring many errors and the prediction result of the solar flux will be inaccurate. Moreover, the distance between the heliostat and receiver plane is also required to be large enough. Furthermore, the projection method must be applied to bring many more errors.

The majority of the heliostats in the current tower solar system apply the mirror with a reflecting surface on the back of the glass. Experiments show that the reflected rays are mainly divided into three parts: the rays directly reflected from the glass surface, the main rays reflected from the metal reflecting surface [11], and part of the main rays reflected off the upper glass surface and then reflected back off the metal reflecting surface. In current studies of solar tower systems, they are all treated as the same reflected beam, which is different from the actual situation.

In this paper, under the only assumption of an approximate description of the sun shape, a direct integration method was applied to deduce an integration equation to compute the solar flux reflected from the heliostat at any point on the receiver plane, considering the influence of the optical error of the heliostat and multiple reflections by the two planes. An elliptic Gaussian model was used to describe the optical error of the heliostat. A numerical code was developed to calculate a rectangular spherical heliostat's flux density distribution and intercept factor based on this model. Compared with the experimental data of ten heliostats, the applicability and accuracy of the model were verified.

## 2. Methodology

### 2.1. The Energy Density Ratios of the Reflected Ray

Solar rays from various positions will reach different positions on the receiver surface after being reflected by a certain point of the heliostat. Optical errors at different points of the heliostat will cause the reflected ray to deviate from the ideal direction, as shown in Figure 1. We calculated the flux density contribution of the reflected solar ray to a certain point on the receiver surface by integrating all of the reflection points on the heliostat [9]. Therefore, the distribution of the flux density on the whole receiver surface could be computed.

The rays reflected by the heliostat were divided into three main parts [11]: $I_1(4\%)$, $I_2(89.5\%)$, and $I_3(3.5\%)$, as shown in Figure 2. The incident rays first reach the upper

surface of the glass, and part $I_1$ is directly reflected, accounting for approximately 5%. The rest of the rays are transmitted through glass, reflected by the metal reflecting surface, and transmitted out of the upper surface. These are the main reflected rays, namely part $I_2$, accounting for approximately 91.5% when the incidence angle is small. The remaining part is reflected again from the upper surface, and again from the reflective metal surface. When it reaches the upper surface, it transmits out part $I_3$, accounting for approximately 3%. The rest of the rays still travel inside the glass, but the fraction is negligible.

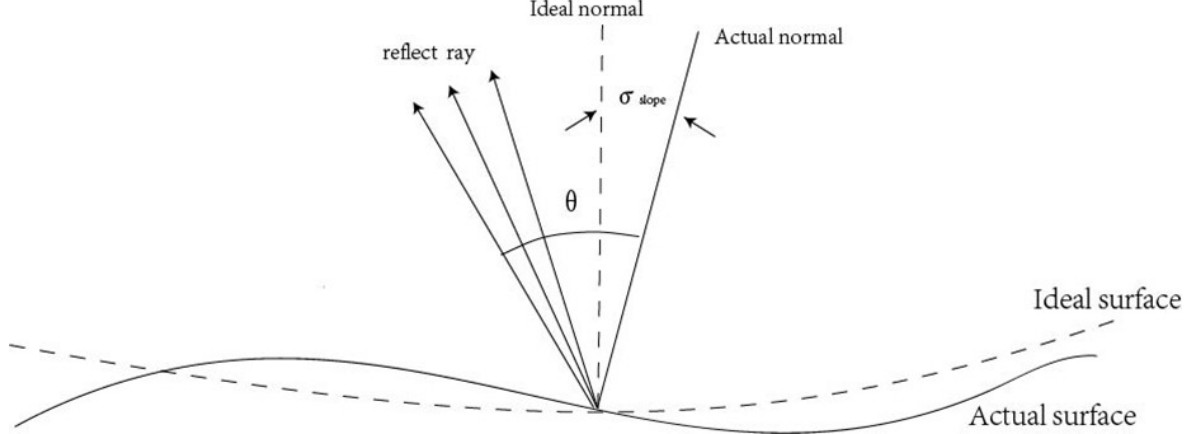

**Figure 1.** Slope error $\sigma_{slope}$ and reflection cone $\theta$.

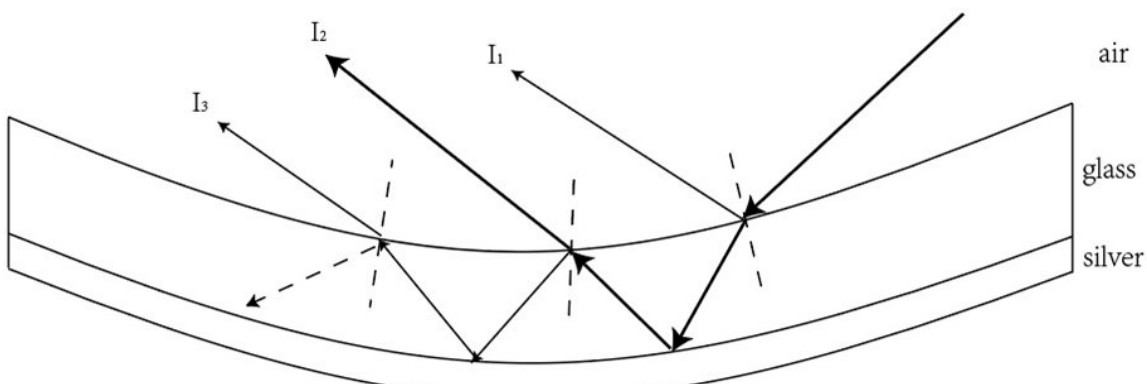

**Figure 2.** Composition of reflected rays.

The focuses of the three parts of the beams are different, but the part $I_2$ mainly determines the distribution of the flux density on the receiver plane, so, in practice, the rays of part $I_2$ are aligned with the center of the receiver plane. When the distance between the receiver plane and the heliostat is large enough, the difference in the focal positions of the three parts can be ignored. Meanwhile, the contribution of part $I_1$ and part $I_3$ to the flux density is not tiny. When a more accurate flux density distribution is required, the optical errors of these two parts should be taken into account, and the flux density distribution is different from that of part $I_2$. In particular, part $I_1$ and $I_3$ are not taken into account when calculating the intercept factor, and the curvature of the intercept curve will differ significantly from the actual situation, especially when the intercept factor reaches 90%.

The proportion of $I_1$, $I_2$, and $I_3$ is related to the incident angle. There are also polarization changes in reflection and refraction on the glass surface. We first decomposed the amplitude vector of the ray into two directions. The one perpendicular to the vibration of the incident plane is called the S component, and the one parallel to the vibration of the incident plane is called the P component. They have identical parts in solar rays.

The incident angle is $i_1$ and the ratios of the refractive index of the two sides of the reflected glass interface are $n_1$ and $n_2$, respectively. According to the refraction law, the refraction angle inside the glass can be calculated as $i_2$.

According to Fresnel's equations [12], the reflectivity of the reflected ray in the $P$ and $S$ components can be calculated:

$$R_p = |r_p|^2 = \frac{tan^2(i_1 - i_2)}{tan^2(i_1 + i_2)} \tag{1}$$

$$R_s = |r_s|^2 = \frac{sin^2(i_1 - i_2)}{sin^2(i_1 + i_2)} \tag{2}$$

Similarly, the transmissivity of the refracted ray in the $P$ and $S$ components can be calculated:

$$T_p = \frac{n_2 cosi_2}{n_1 cosi_1}|t_p| = \frac{sini_1}{sini_2} \times \frac{cosi_2}{cosi_1} \times (\frac{2sini_2 cosi_1}{sin(i_1 + i_2)cos(i_1 - i_2)})^2 \tag{3}$$

$$T_s = \frac{n_2 cosi_2}{n_1 cosi_1}|t_s| = \frac{sini_1}{sini_2} \times \frac{cosi_2}{cosi_1} \times (\frac{2sini_2 cosi_1}{sin(i_1 + i_2)})^2 \tag{4}$$

As light passes through the metal reflecting surface, namely the silver plating layer, it is refracted and reflected as it reacts with the silver. Part of the refraction into the silver plating layer is equivalent to loss:

$$R_s^{silver} = \frac{(n_s - cosi_s)^2 + n_s^2 \times k^2}{(n_s + cosi_s)^2 + n_s^2 \times k^2} \tag{5}$$

$$R_p^{silver} = \frac{(n_s - \frac{1}{cosi_s})^2 + n_s^2 \times k^2}{(n_s + \frac{1}{cosi_s})^2 + n_s^2 \times k^2} \tag{6}$$

$R_s^{silver}$ and $R_p^{silver}$ are the reflectivity of light on the silver surface in the $P$ and $S$ components. $i_3$ is the incident angle reaching the silver surface. $n_3$ is the ratio of the refractive index of the silver plating layer relative to the glass, and, here, it was equal to 0.051585. $k$ is the dielectric constant, and, here, it was equal to 3.9046. The wavelength of the incident ray was 587.6 nm [13].

As light passes through the glass, some of it is absorbed by the glass. Heliostats usually use ultra-clear glass. According to the experimental results, approximately 1% [14] of light is absorbed when it passes through the glass perpendicularly, mainly absorbed by impurities in the mirror.

The incident ray is natural light, and the light intensity is $I_0$, which has the same amplitude in the $P$ and $S$ components. Thus, the light intensity in the $P$ and $S$ components is equal.

By solving the energy density of the reflected and refracted ray at the interface boundary each time, the energy density of the three parts of rays reflected off the glass surface can be obtained. Finally, the proportion of three parts of the reflected ray can be calculated.

$$I_1 = \frac{1}{2} \times I_0 \times (T_p \times R_s) \tag{7}$$

$$I_2 = \frac{1}{2} \times I_0 \times (T_p \times T_p' \times R_p^{silver} + T_s \times T_s' \times R_p^{silver}) \times (1 - \alpha)^2 \tag{8}$$

$$I_3 = \frac{1}{2} \times I_0 \times (T_p \times R_p' \times T_p' \times (R_p^{silver})^2 + T_s \times R_s' \times T_s' \times (R_p^{silver})^2) \times (1 - \alpha)^4 \tag{9}$$

where $R_p$ and $R_s$ are the reflectivity of the first reflections in the $P$ and $S$ components, respectively. $R_p'$ and $R_s'$ are the reflectivity of the second reflections in the $P$ and $S$ components, respectively. $T_p$ and $T_s$ are the transmissivities of the transmission into the glass in

the $P$ and $S$ components, respectively. $T'_p$ and $T'_s$ are the transmissivities of the transmission out of the glass in the $P$ and $S$ components, respectively. $\alpha$ is the absorption ratio of glass to light, which is calculated by:

$$\alpha = 1 - exp(-\varepsilon \times \frac{d}{cosi_2}) \tag{10}$$

where $d$ is the thickness of the glass, and $\varepsilon$ is the absorption coefficient, which should be determined from the test data [14] for the specific mirror.

Finally, the energy density ratios of the three parts can be calculated:

$$\delta = \frac{I_i}{I_1 + I_2 + I_3}(i = 1, 2, 3) \tag{11}$$

### 2.2. Solution of Flux Density on Receiver Plane

Although the energy density distribution formed by the three parts of the reflected ray is different, the calculation process of the flux density of each part is similar. It only needs to calculate the flux density on the receiver plane of a single part and then multiply it by the energy density ratios to obtain the flux density of the three parts. Finally, we can calculate the flux density distribution on the whole receiver plane.

$$F_{total} = \sum_{i=1}^{3} \delta_i \times F_i(i = 1, 2, 3) \tag{12}$$

where $F_i$ is the flux density of a single part.

For a solar tower system, it is assumed that a receiver plane is a square plane and the heliostat is a rectangular spherical mirror, as shown in Figure 3. Suppose that the heliostat reflector is $S_1$, the receiver plane is $S_2$, $n_1$ and $n_2$ are normals of the cell surfaces on the heliostat and the receiver, $O_1$ and $O_2$ are the points on the cell surfaces on the heliostat and the receiver, and $\beta_1$ and $\beta_2$ are the included angles between the line $O_1O_2$ and normals of each cell surface. For a single part, the analytic formula of the flux density at point $O_2$ in the receiver surface is [15]:

$$F = \iint_{S_1} \frac{IBcos\beta_1 cos\beta_2}{r^2} dS_1 \tag{13}$$

where $r = |O_1O_2|$ is the distance between the reflection point and the receiver point, $I$ is the total intensity of the reflected ray of a single part, and $B$ is the intensity distribution of reflected rays, obtained from the convolution calculation of the sun shape and optical error.

To calculate the intensity distribution of the reflected ray, we need to determine the solar intensity distribution firstly, which is described by a circular Gaussian function. An approximate description of the intensity of sunlight is the only assumption in this paper. Due to many factors, such as sun position and atmospheric absorption, the solar intensity changes from time to time. However, the standard deviation of the solar intensity distribution $\sigma_{sun}$ can be determined by direct solar irradiation $I_D$ [8]:

$$\sigma_{sun} = \frac{[3.7648 - 3.8413(I_D - 1) + 1.5923 \times 10^2(I_D - 1)^2]}{\sqrt{2}} \tag{14}$$

The optical error of the reflected ray must then be determined. This paper adopted the elliptic Gaussian function to describe the slope error on a heliostat. It is necessary to consider that rays will be reflected by the upper and lower surfaces of the mirror. The equations for calculating optical errors are also different for each part of the reflected ray.

The optical error of each point on the heliostat is different [9,16]. To improve the calculation speed, we can integrate the whole heliostat and replace the optical error of each point with the average optical error. To compare the two calculation results, we calculated the difference with ten heliostats in Section 3.2. From the results, the average difference

between the two is less than 0.05%. Therefore, in the subsequent calculation of the optical error, the optical error of each point was no longer considered separately but was only replaced by the average slope error of the surface.

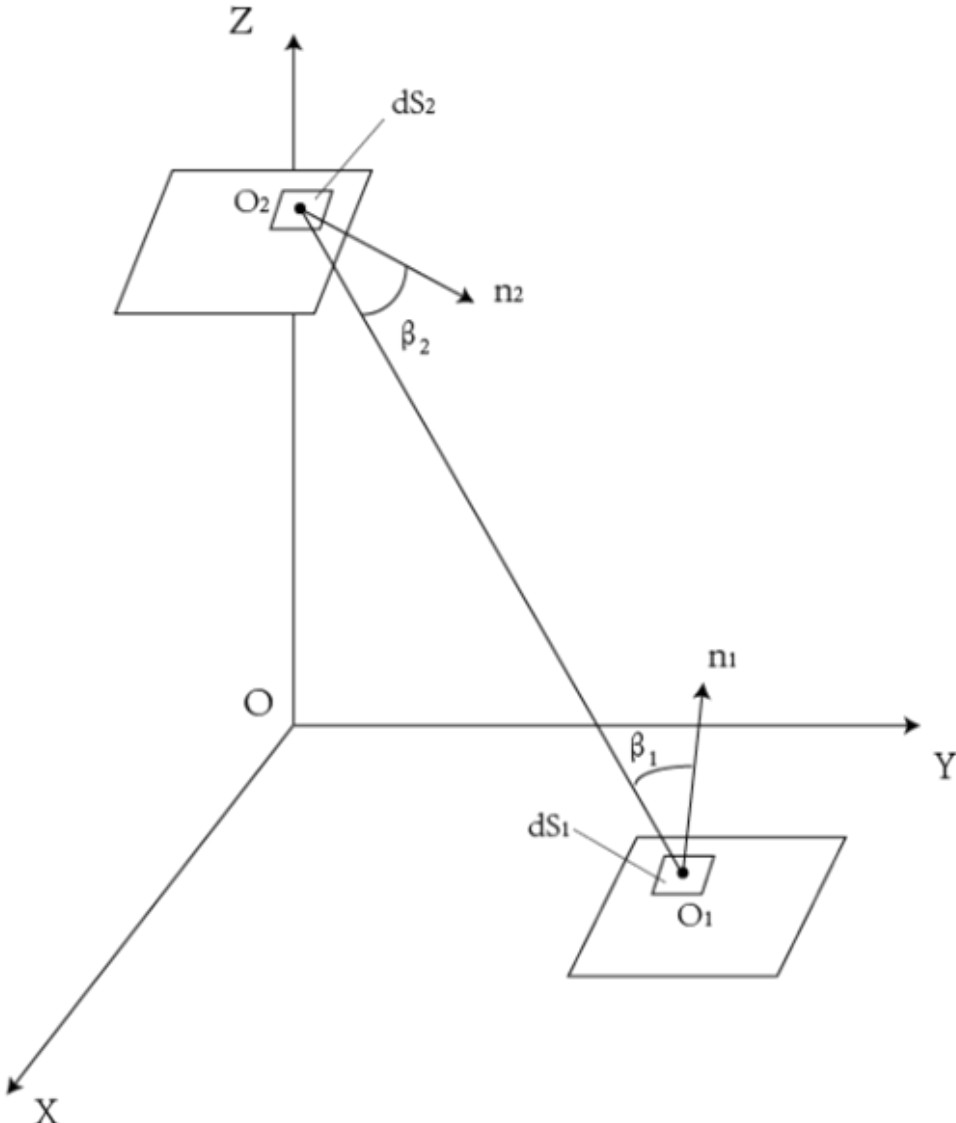

**Figure 3.** X axis points due east, Y axis points due north, and Z axis points zenith.

For the first part of the reflected ray $I_1$, the upper surface of the glass directly reflects it, so the average optical errors were calculated as follows:

$$\sigma_{sx}^2 = 4\sigma_{slopex}^2 + \left(\frac{2tan^2\lambda cos\beta sin\beta}{1 + tan^2\lambda cos^2\beta}\sigma_{slopey}\right)^2 \tag{15}$$

$$\sigma_{sy}^2 = 4\sigma_{slopex}^2 + \left(\frac{2tan^2\lambda cos\beta sin\beta}{1 + tan^2\lambda cos^2\beta}\sigma_{slopex}\right)^2 \tag{16}$$

where $\sigma_{slopex}$ and $\sigma_{slopey}$ are slope errors in the X direction and Y direction, respectively.

For the second part of the reflected ray $I_2$, it will be refracted by the lower surface and reflected by the upper surface. Given mirrors' relatively mature manufacturing technology, the slope errors on the upper and lower surfaces can be considered equal.

Therefore, the standard deviation of the average error in two directions was calculated as follows:

$$
\sigma_{sx}^2 = (1 - \frac{cos\lambda}{\sqrt{n_0^2 - sin^2\lambda}})^2[\sigma_{slopex}^2 + (\frac{2n_0^2 - sin^2\lambda}{2n_0\sqrt{n_0^2 - sin^2\lambda}} - 1)\sigma_{slopey}^2] + 4\sigma_{slopex}^2 +
$$
$$
(\frac{4n_0^2 - 2sin^2\lambda}{n_0\sqrt{n_0^2 - sin^2\lambda}} - 1)\sigma_{slopey}^2 + (1 - \frac{\sqrt{n_0^2 - sin^2\lambda}}{cos\lambda})^2(\sigma_{slopex}^2 + \frac{(1 - cos\lambda)^2}{2cos\lambda}\sigma_{slopey}^2)
$$

$$(17)$$

$$
\sigma_{sy}^2 = (1 - \frac{cos\lambda}{\sqrt{n_0^2 - sin^2\lambda}})^2[\sigma_{slopey}^2 + (\frac{2n_0^2 - sin^2\lambda}{2n_0\sqrt{n_0^2 - sin^2\lambda}} - 1)\sigma_{slopex}^2] + 4\sigma_{slopey}^2 +
$$
$$
(\frac{4n_0^2 - 2sin^2\lambda}{n_0\sqrt{n_0^2 - sin^2\lambda}} - 1)\sigma_{slopex}^2 + (1 - \frac{\sqrt{n_0^2 - sin^2\lambda}}{cos\lambda})^2(\sigma_{slopey}^2 + \frac{(1 - cos\lambda)^2}{2cos\lambda}\sigma_{slopex}^2)
$$

$$(18)$$

where $n_0$ is the refractive index ratio of glass to air, constant at 1.51 in general, and $\lambda$ is the incident angle of the heliostat reflection point, respectively.

For the third part of the ray $I_3$, the ray passes through the glass again, and its optical error is two more reflection errors than that of the ray $I_2$:

$$
\sigma_{sx}^2 = (1 - \frac{cos\lambda}{\sqrt{n_0^2 - sin^2\lambda}})^2[\sigma_{slopex}^2 + (\frac{2n_0^2 - sin^2\lambda}{2n_0\sqrt{n_0^2 - sin^2\lambda}} - 1)\sigma_{slopey}^2] + 12\sigma_{slopex}^2 +
$$
$$
(\frac{4n_0^2 - 2sin^2\lambda}{n_0\sqrt{n_0^2 - sin^2\lambda}} - 1)\sigma_{slopey}^2 + (1 - \frac{\sqrt{n_0^2 - sin^2\lambda}}{cos\lambda})^2(\sigma_{slopex}^2 + \frac{(1 - cos\lambda)^2}{2cos\lambda}\sigma_{slopey}^2)
$$

$$(19)$$

$$
\sigma_{sy}^2 = (1 - \frac{cos\lambda}{\sqrt{n_0^2 - sin^2\lambda}})^2[\sigma_{slopey}^2 + (\frac{2n_0^2 - sin^2\lambda}{2n_0\sqrt{n_0^2 - sin^2\lambda}} - 1)\sigma_{slopex}^2] + 12\sigma_{slopey}^2 +
$$
$$
(\frac{4n_0^2 - 2sin^2\lambda}{n_0\sqrt{n_0^2 - sin^2\lambda}} - 1)\sigma_{slopex}^2 + (1 - \frac{\sqrt{n_0^2 - sin^2\lambda}}{cos\lambda})^2(\sigma_{slopey}^2 + \frac{(1 - cos\lambda)^2}{2cos\lambda}\sigma_{slopex}^2)
$$

$$(20)$$

The total error of the reflected ray at each point of the heliostat was calculated by convolution of three Gaussian function errors as follows:

$$
\sigma_x^2 = \sigma_{sun}^2 + \sigma_{sx}^2 + \sigma_{trx}^2 \tag{21}
$$

$$
\sigma_y^2 = \sigma_{sun}^2 + \sigma_{sy}^2 + \sigma_{try}^2 \tag{22}
$$

where $\sigma_{trx}$ and $\sigma_{try}$ are the tracking errors in the X direction and Y direction. The tracking errors of all models used in this paper were 0.

In the elliptic Gaussian model, the distribution of reflected light intensity can be calculated as:

$$
B(\theta_x, \theta_y) = \frac{1}{2\pi\sigma_x\sigma_y} \exp[-\frac{1}{2}(\frac{\theta_x^2}{\sigma_x^2} + \frac{\theta_y^2}{\sigma_y^2})] \tag{23}
$$

where $\theta_x$ is the angle between $O_1O_2$ in the X direction and the reflected ray from the center of each cell on the heliostat, and $\theta_y$ is the angle between $O_1O_2$ in the Y direction and the reflected ray from the center of each cell on the heliostat. $\sigma_x$ and $\sigma_y$ are the total errors in the X direction and Y direction, respectively.

In this paper, based on the elliptic Gaussian model, we simplified the calculation process of an optical error and built a circular Gaussian model, which was applied to verify the model's accuracy by comparing it with SolTrace [17]. It can also be applied to quickly

compute a flux density distribution when the accuracy requirement is not high. In the circular Gaussian model, the reflected ray was no longer divided into three parts and was only calculated as the same reflected ray. The slope error $\sigma_{slope}$ was simplified to radial, and the tracking error $\sigma_{tr}$ was taken into account, so the total error was calculated as [18]:

$$\sigma^2 = \sigma_{sun}^2 + \sigma_{slope}^2 + \sigma_{tr}^2 \tag{24}$$

In the later model verification, the optical error in SolTrace will be consistent with that in the circular Gaussian model and calculated by the least square method.

Therefore, the intensity distribution function of the reflected ray in the circular Gaussian model can be expressed as:

$$B(\theta) = \frac{1}{2\pi\sigma^2} \exp(-\frac{\theta^2}{2\sigma^2}) \tag{25}$$

$\theta$ is the angle between $O_1O_2$ and the reflected ray from the center of each cell on the heliostat, and $\sigma$ is the total error.

Therefore, for a single part of a reflected ray, the flux density distribution function that the heliostat devotes to a specific point on the receiver plane can be transformed into:

Circular Gaussian model:

$$F = \iint_{S_1} \frac{I cos\beta_1 cos\beta_2 \exp(-\frac{\theta^2}{2\sigma^2})}{2\pi\sigma^2 r^2} dS_1 (0 \le \beta_1, \beta_2 \le \frac{\pi}{2}) \tag{26}$$

Elliptic Gaussian model:

$$F = \iint_{S_1} \frac{I cos\beta_1 cos\beta_2 \exp[-\frac{1}{2}(\frac{\theta_x^2}{\sigma_x^2} + \frac{\theta_y^2}{\sigma_y^2})]}{2\pi\sigma_x\sigma_y r^2} dS_1 (0 \le \beta_1, \beta_2 \le \frac{\pi}{2}) \tag{27}$$

### 2.3. Flux Density Computation Method

When considering the intensity of the reflected solar ray, we first divided the heliostat into smaller grids, and the center point of each grid was used to represent the grids. Then, we solved the flux density contribution of all reflected rays to a certain point on the receiver plane. Finally, we integrated these contributions to compute the flux density value of a specific point on the receiver plane [19].

The actual calculation process of a specific point is mainly divided into four steps:

i　　Location determination.
ii　　Meshing process and coordinate system rotation.
iii　　Calculation of geometrical optics.
iv　　Solution of the flux density at a specific point.

The calculation details are shown in Figure 4. Further, the flux density of other grid points was calculated according to the above method. As shown in Appendix A, the actual calculation process is very complicated, so we developed a numerical code to help us complete the analysis and calculation.

To verify the reliability of the proposed method, we calculated the root mean square error (RMSE) of the flux density and intercept factor between the models and experimental data [19]:

$$RMSE = \sqrt{\frac{\sum_1^u \sum_1^v [F_1(u,v) - F_2(u,v)]^2}{uv - 1}} \tag{28}$$

where $F_1(u,v)$ and $F_2(u,v)$ are the flux density calculated by the model and measured data, respectively. This formula can also calculate the root mean square error between two different models.

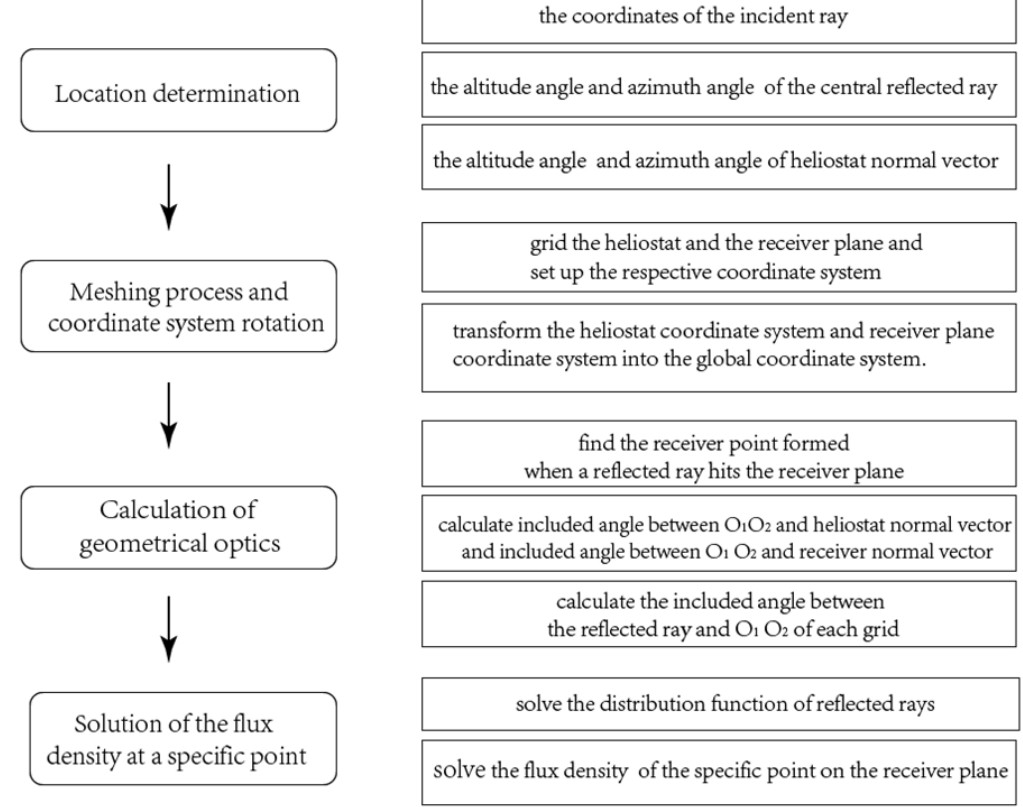

**Figure 4.** Solution of the flux density at a specific point.

## 3. Model Validation

### 3.1. Comparison with SolTrace

SolTrace [17] is a software that simulates the real optical action process based on the traditional ray-tracing method, developed by The National Renewable Energy Laboratory (NREL). In SolTrace, the flux density value calculated from five million random rays is accurate enough, so we set the number of rays to five million in the subsequent calculation using SolTrace.

SolTrace can only be used to calculate the solar intensity radial distribution model. Therefore, we can compare the results of SolTrace with that of the circular Gaussian model, which were used to verify the calculation results of the circular Gaussian model.

The origin of the coordinate system is located in the center of the bottom of the tower. The positive direction of the Y-axis is due north, the positive direction of the X-axis is due east, and the positive direction of the Z-axis is the zenith. The tower height is 120 m, the receiver is placed perpendicular to the ground, the solar altitude angle is $45°$, the solar azimuth angle is $0°$, and the solar light intensity is $1 \text{ kW/m}^2$. All parameters of the heliostat are shown in Table 1. The slope error in the circular Gaussian model is 1.7 mrad, consistent with the slope error setting in SolTrace. After setting each parameter, we calculated the flux density distribution of the rectangular heliostat using the circular Gaussian model and SolTrace, respectively.

**Table 1.** Heliostat parameters.

| $X$ (m) | $Y$ (m) | $Z$ (m) | $f_0$ (m) | Width | Length | $\cos\lambda$ | $\sigma_{sun}$ (mrad) | $\sigma_{tr}$ (mrad) |
|---|---|---|---|---|---|---|---|---|
| 300 | 500 | 0 | 595.3 | 6 | 9 | 0.8022 | 2.51 | 0 |

Figure 5a shows the distribution of the flux density on the receiver surface computed by SolTrace and the circular Gaussian model. As SolTrace can only apply the circular Gaussian model, here, we also used it in our model for comparison. Since the reflected

ray is at an acute large incident angle to the receiver surface, the distribution of the flux density on the receiver surface is elliptical, and the average absolute difference computed by the two methods is 0.64%. Figure 5b shows the intercept factor calculated by SolTrace and the circular Gaussian model. On the whole, the intercept factor computed by SolTrace is slightly larger than that. The figures show that the results computed by the circular Gaussian model and SolTrace have a good consistency.

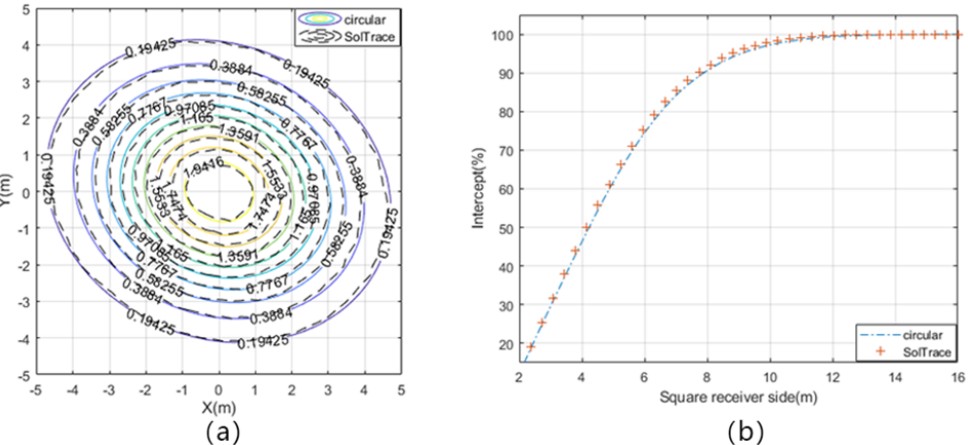

**Figure 5.** (**a**) Flux density distribution computed by the circular Gaussian model and SolTrace (kW/m$^2$), (**b**) intercepts computed by the circular Gaussian model and SolTrace.

### 3.2. Verification with Experimental Data

During July, Collado carried out an experiment in the tower heliostat system [20]. He recorded the experimental data, which included the measured solar flux contours and intercept factors of the heliostat in ten different positions on the receiver plane. The ten heliostats measure 6.6778 m in width and 6.819 m in length. The spherical heliostat's total mirror area is 39.9126 m$^2$. Figure 6 depicts the position distribution of heliostats.

The experimental results of the flux density distribution and intercept factor were compared to SolTrace, the elliptic Gaussian, and the circular Gaussian models. The slope errors of all models were calculated using the least square method. Meanwhile, the reflected light intensity distribution in SolTrace is consistent with a circular Gaussian model. Because the circular Gaussian model results are not significantly different from those of SolTrace, they were only recorded as errors in the tables.

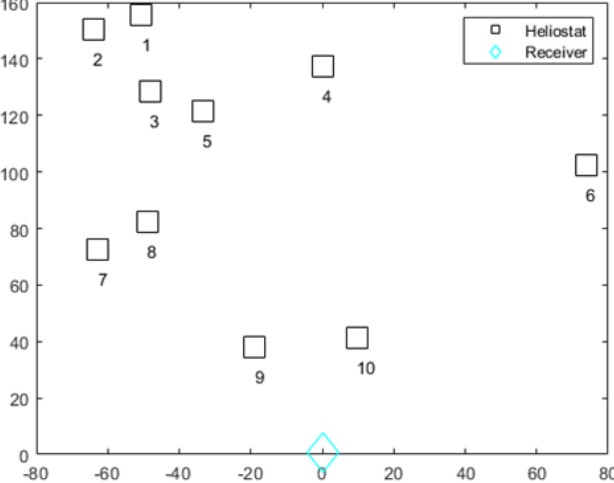

**Figure 6.** Distribution of ten heliostats and receiver.

The elliptic Gaussian model must first calculate the energy density ratios of ten heliostats' reflected rays. Given the incident angles of the ten heliostats, and after accounting for reflection

and refraction at the glass interface and light energy loss at the silver plating layer, the energy density ratios of the reflected rays of each heliostat were calculated, as shown in Table 2. The energy density ratios of $I_1$, $I_2$, and $I_3$ were approximately 5%, 91.5%, and 3.5%, respectively.

**Table 2.** Parameters of ten heliostats.

| Heliostats | Incidence (Rad) | $\delta_1$ | $\delta_2$ | $\delta_3$ |
|---|---|---|---|---|
| 1 | 0.8188 | 5.08% | 91.50% | 3.42% |
| 2 | 0.8477 | 4.96% | 91.65% | 3.39% |
| 3 | 0.8253 | 5.05% | 91.53% | 3.42% |
| 4 | 0.8510 | 4.95% | 91.67% | 3.38% |
| 5 | 0.8327 | 5.02% | 91.58% | 3.40% |
| 6 | 0.8421 | 4.98% | 91.63% | 3.39% |
| 7 | 0.8922 | 4.83% | 91.82% | 3.35% |
| 8 | 0.8740 | 4.88% | 91.76% | 3.36% |
| 9 | 0.9484 | 4.75% | 91.92% | 3.33% |
| 10 | 0.9513 | 4.74% | 91.93% | 3.33% |

The slope errors of all models were computed by the least square method. Table 3 shows the optical errors of the three parts of the elliptic Gaussian model, SolTrace, and circular Gaussian models. Meanwhile, the optical errors of SolTrace and the circular Gaussian model refer to the average slope error of the mirror surface.

**Table 3.** Optical errors of ten heliostats calculated by different models (unit: mrad).

| Heliostats | $\sigma-$ Elliptical-$I_1-$ $x$ | $\sigma-$ Elliptical-$I_1-$ $y$ | $\sigma-$ Elliptical-$I_2-$ $x$ | $\sigma-$ Elliptical-$I_2-$ $y$ | $\sigma-$ Elliptical-$I_3-$ $x$ | $\sigma-$ Elliptical-$I_3-$ $y$ | $\sigma-$ SolTrace and Circular Optical |
|---|---|---|---|---|---|---|---|
| 1 | 1.93 | 1.56 | 2.27 | 2.01 | 3.72 | 3.28 | 2.10 |
| 2 | 1.79 | 2.58 | 2.39 | 2.94 | 3.92 | 4.85 | 2.73 |
| 3 | 2.37 | 0.26 | 2.56 | 1.36 | 4.22 | 2.18 | 2.07 |
| 4 | 2.85 | 1.40 | 3.15 | 2.18 | 5.20 | 3.55 | 2.58 |
| 5 | 2.56 | 1.30 | 2.85 | 2.01 | 4.70 | 3.26 | 2.46 |
| 6 | 1.61 | 1.76 | 1.99 | 2.01 | 3.27 | 3.45 | 2.36 |
| 7 | 0.67 | 3.62 | 2.09 | 3.87 | 3.42 | 6.43 | 3.26 |
| 8 | 1.76 | 2.11 | 2.20 | 2.45 | 3.63 | 4.05 | 2.62 |
| 9 | 6.49 | 3.65 | 7.09 | 5.16 | 11.86 | 8.57 | 3.84 |
| 10 | 7.07 | 3.67 | 3.68 | 5.37 | 12.85 | 8.93 | 5.54 |

Figures 7–16 show the flux density distribution and intercept factor of ten heliostats measured by the experiment and computed using the circular Gaussian model and SolTrace. The elliptic Gaussian model in this paper is superior to SolTrace in flux density distribution and intercept factor calculation.

The absolute average difference in distribution of the flux density and intercept factor between the elliptic Gaussian model and experimental data are shown in Tables 4 and 5. We compared the flux density and intercept factor calculated by multiple models, and the calculation results of SolTrace and the circular Gaussian model are also shown in the tables.

In terms of the flux density distribution, the result indicates that the average absolute difference between the elliptic Gaussian and measured data is 2.83%. The average absolute difference between the circular Gaussian and measured data is 4.40%, and the average absolute difference between SolTrace and measured data is 3.50%. Therefore, the average absolute difference between the elliptic Gaussian model and measured SolTrace data is the smallest. The calculation result of heliostat 10 is relatively the best, with a 1.91% smaller average difference than that of SolTrace.

The flux density distribution calculated by the elliptic Gaussian model and SolTrace was compared using Heliostat #10. The elliptic Gaussian model simulates the experimental data's flux density contour well, as shown in Figure 16a, and the long and short axes are consistent. However, the SolTrace-calculated results deviated from the measured contour

and could not adequately fit the flux density distribution. In terms of data, the average absolute difference in flux density distribution between the elliptic Gaussian model and the experimental data is 3.38%, which is significantly smaller than the 4.43% of SolTrace and the 5.97% of circular Gaussian.

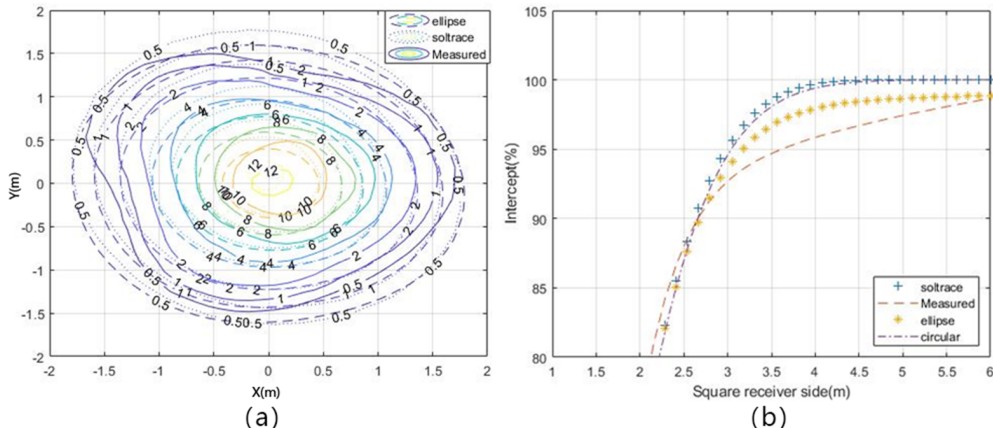

**Figure 7.** (**a**) Contours of the measured and computed flux distribution (kW/m$^2$): Heliostat #1, (**b**) measured and computed intercepts vs. the side length of a square receiver: Heliostat #1 .

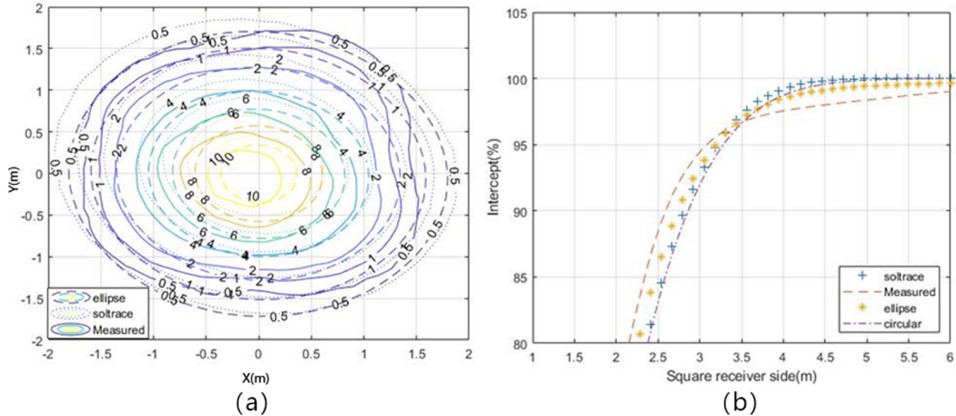

**Figure 8.** (**a**) Contours of the measured and computed flux distribution (kW/m$^2$): Heliostat #2, (**b**) measured and computed intercepts vs. the side length of a square receiver: Heliostat #2 .

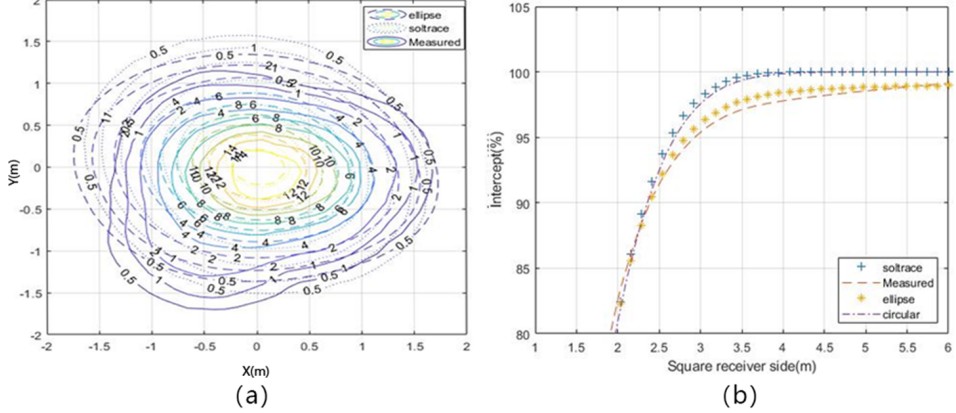

**Figure 9.** (**a**) Contours of the measured and computed flux distribution (kW/m$^2$): Heliostat #3, (**b**) measured and computed intercepts vs. the side length of a square receiver: Heliostat #3 .

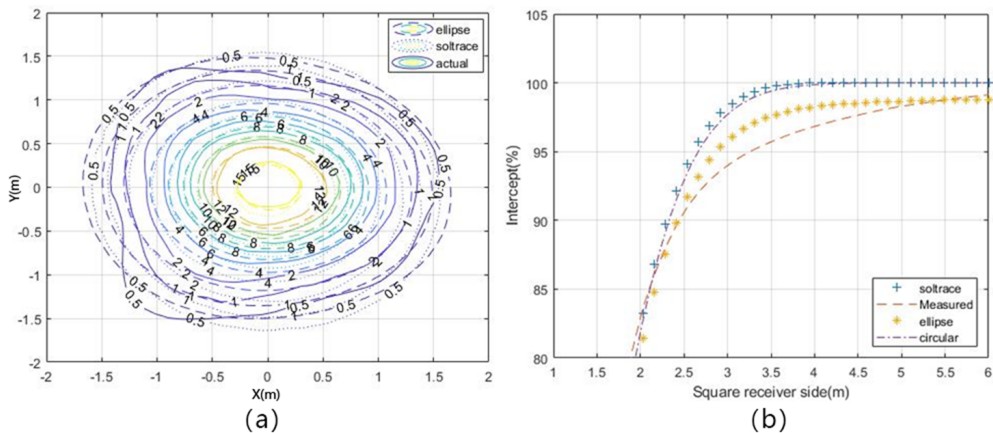

**Figure 10.** (**a**) Contours of the measured and computed flux distribution (kW/m$^2$): Heliostat #4, (**b**) measured and computed intercepts vs. the side length of a square receiver: Heliostat #4 .

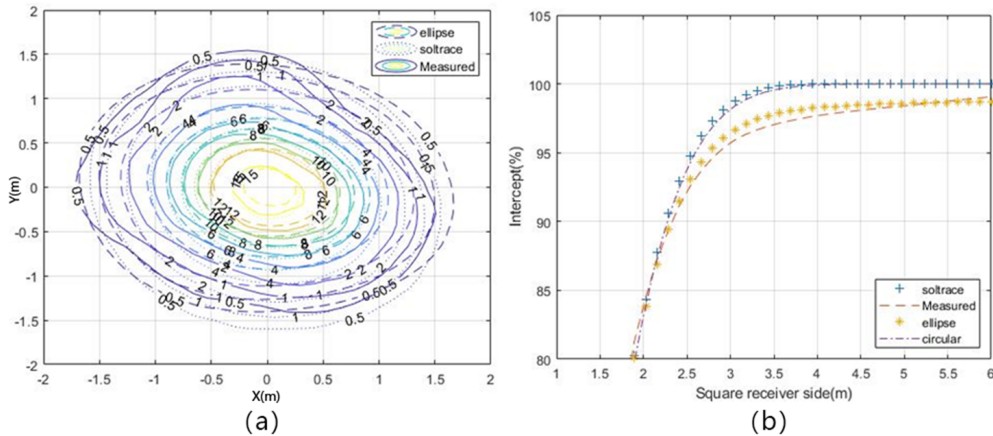

**Figure 11.** (**a**) Contours of the measured and computed flux distribution (kW/m$^2$): Heliostat #5, (**b**) measured and computed intercepts vs. the side length of a square receiver: Heliostat #5 .

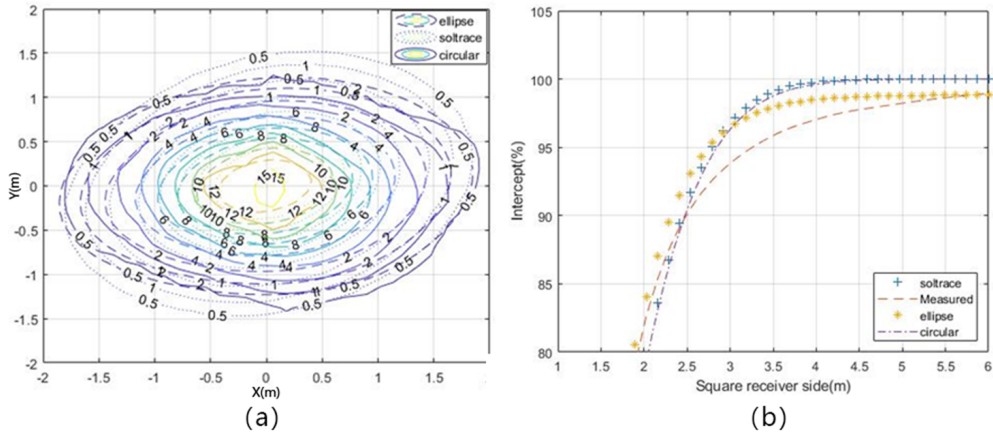

**Figure 12.** (**a**) Contours of the measured and computed flux distribution (kW/m$^2$): Heliostat #6, (**b**) measured and computed intercepts vs. the side length of a square receiver: Heliostat #6 .

In terms of the intercept, the elliptic Gaussian model and circular Gaussian model proposed in this paper are consistent with the intercept factor calculated by SolTrace. The result of intercept factors suggests that the absolute average difference in the elliptic Gaussian and measured data is 1.13%, the absolute average difference in the circular

Gaussian and measured data is 2.28%, and the absolute average difference in SolTrace and measured data is 2.36%. In most cases, the intercept factor of the elliptic Gaussian model is much better than that of SolTrace.

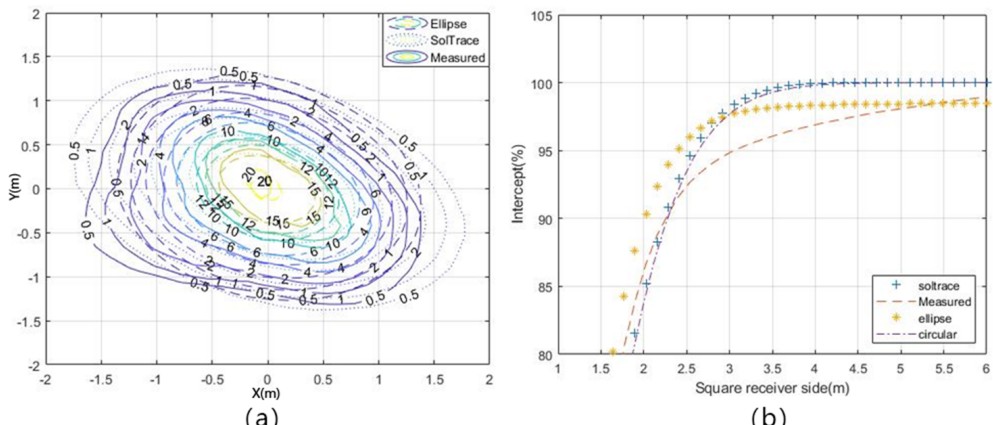

**Figure 13.** (**a**) Contours of the measured and computed flux distribution (kW/m$^2$): Heliostat #7, (**b**) measured and computed intercepts vs. the side length of a square receiver: Heliostat #7 .

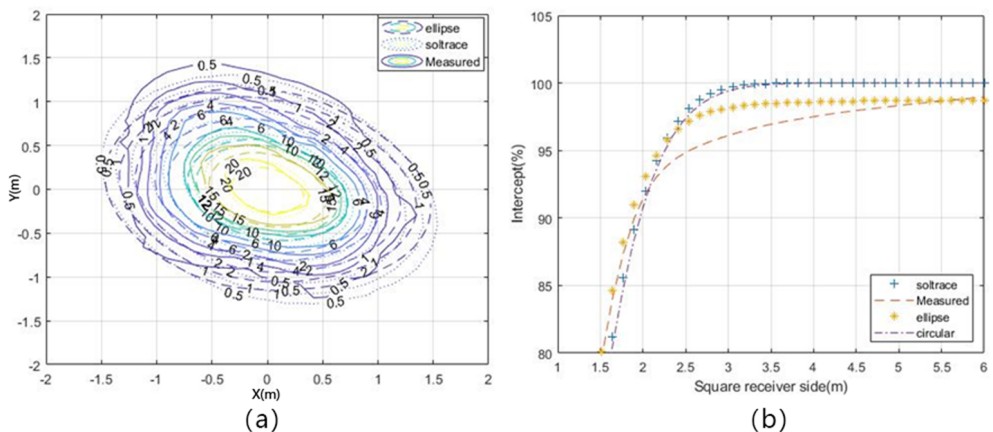

**Figure 14.** (**a**) Contours of the measured and computed flux distribution (kW/m$^2$): Heliostat #8, (**b**) measured and computed intercepts vs. the side length of a square receiver: Heliostat #8 .

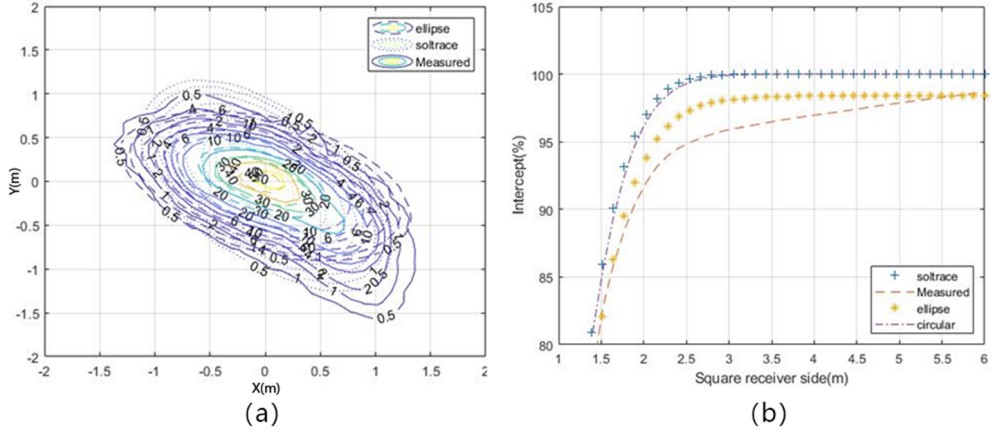

**Figure 15.** (**a**) Contours of the measured and computed flux distribution (kW/m$^2$): Heliostat #9, (**b**) measured and computed intercepts vs. the side length of a square receiver: Heliostat #9 .

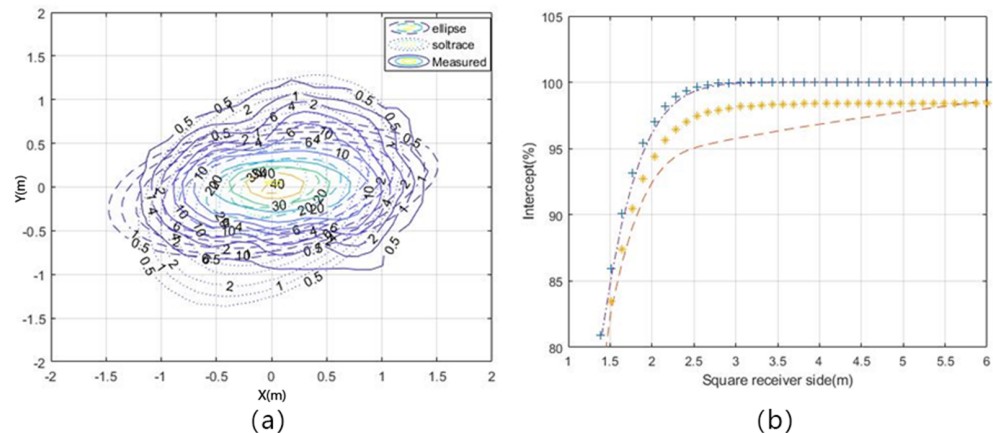

**Figure 16.** (**a**) Contours of the measured and computed flux distribution (kW/m$^2$): Heliostat #10, (**b**) measured and computed intercepts vs. the side length of a square receiver: Heliostat #10 .

**Table 4.** The absolute average difference in flux density of ten heliostats.

| Heliostats | Elliptical | SolTrace | Circular |
|---|---|---|---|
| 1 | 3.32 | 3.87 | 4.04 |
| 2 | 2.86 | 3.38 | 4.57 |
| 3 | 3.64 | 4.41 | 5.26 |
| 4 | 2.40 | 2.87 | 3.34 |
| 5 | 2.33 | 2.89 | 3.13 |
| 6 | 2.60 | 3.72 | 5.20 |
| 7 | 2.87 | 3.21 | 3.50 |
| 8 | 2.40 | 2.88 | 3.72 |
| 9 | 2.50 | 3.30 | 5.28 |
| 10 | 3.38 | 4.43 | 5.97 |

**Table 5.** The absolute average difference in intercept factor of ten heliostats.

| Heliostats | Elliptical | SolTrace | Circular |
|---|---|---|---|
| 1 | 1.27 | 2.64 | 2.50 |
| 2 | 1.26 | 2.09 | 2.22 |
| 3 | 0.46 | 1.76 | 1.64 |
| 4 | 1.21 | 2.54 | 2.35 |
| 5 | 0.49 | 1.90 | 1.78 |
| 6 | 1.26 | 2.19 | 2.07 |
| 7 | 1.75 | 2.36 | 2.27 |
| 8 | 0.94 | 2.25 | 2.30 |
| 9 | 1.30 | 2.90 | 2.82 |
| 10 | 1.38 | 2.95 | 2.82 |

Taking Heliostat #5 as an example, the intercept factor obtained by experimental data, the elliptic Gaussian model, circular Gaussian model, and SolTrace are shown in Figure 11b. The absolute average difference between the intercept factor calculated by the elliptic Gaussian model and the experimental data is only 0.49%. The absolute average difference between the intercept factor calculated by the circular Gaussian model and the experimental data is 3.12%. The absolute average difference in the intercept factor calculated by SolTrace and the experimental data is 3.25%. The intercept factor calculated by the elliptic Gaussian model is, to a large extent, close to the actual situation. Compared with the circular Gaussian model and SolTrace, the method of the reflected ray being divided into $I_1$, $I_2$, and $I_3$ can well explain the curvature changes in the intercept curve. The intercept factors of SolTrace and the circular Gaussian model are both close to 100% when the side length of the receiver plane reaches 6 m, whereas the intercept factor of the elliptic

Gaussian model and experiment data is close to 98%. This is mainly due to the large optical error of reflected rays in part $I_3$, which prevents it from being completely intercepted when it reaches the receiver plane. The circular Gaussian model and SolTrace do not take this situation into account, so the calculated result of the intercept factor is markedly different from the actual situation.

## 4. Analysis and Discussion

In summary, the average difference, minimum difference, and maximum difference in flux density distribution between the elliptic Gaussian model and experimental data are 2.83%, 2.33%, and 3.64%, and the average difference, minimum difference, and maximum difference in intercept factor are 1.13%, 0.46%, and 1.75%. The error of the model arises from the following aspects:

i       Each point on the heliostat is affected by gravity, temperature, and wind. The slope error will change to different degrees, resulting in the irregular flux density distribution of experimental results and errors with model calculation results.

ii      The solar intensity distribution uses the circular Gaussian function, which is different from the actual solar intensity distribution and will bring some errors.

This model considers rays reflected by two planes of the glass mirror, and the focused light spot formed on the receiver plane is divided into three parts, which has been observed experimentally [11]. This is due to the ray reflected by the upper and lower surfaces of the glass. In this paper, when calculating the flux density of the tower heliostat, its influence was also considered, significantly impacting the calculation result. Although the focused spot position of the three reflections differs little, the error transfer effect of the three reflections needs to be calculated separately due to the difference in the number of reflections. The intensity distribution of reflected rays is very different in three parts, resulting in a different flux density distribution and larger light spot dispersion, which is more consistent with the actual situation and introduces fewer calculation errors.

Conventional ray-tracing methods consider the single beam and use the circular Gaussian model to describe the transfer effect of the surface error distribution. The comparison results in this paper show that the average difference between SolTrace and experimental data is 3.50%, and the maximum difference is 4.43%. In addition, even if the Monte Carlo method [21] is used to reduce the amount of calculation, it is still necessary to simulate as many reflected rays of the heliostat as possible to obtain high-precision results. Therefore, the speed is relatively slow, which is not suitable for optimizing flux density calculation. When using an i7-4790 CPU 3.6 GHz personal computer, SolTrace needs 7.125 s to calculate the flux density distribution reflected by a single heliostat. In comparison, the elliptic Gaussian model in this paper only needs 2.211 s to calculate.

Collado et al. [3] used a circular Gaussian function to describe the flux density formed by a heliostat on the image plane. When calculating the intercept factor based on this method, the error can reach up to 9.3%. The convolution integration method is another common method. A circular Gaussian distribution was first used to describe the surface error of a heliostat, but the difference is relatively large compared with experimental data.

In the 1980s, Lipps [7] proposed the method of convolution of the solar brightness distribution and elliptic Gaussian error to calculate and solve the flux density distribution, which requires an interpolation calculation of the data table to be established in advance. In Lipps' paper, HCOEF and KGEN methods were proposed. HCOEF has a fast calculation speed, and the calculation time is approximately 0.27 s. However, when the slope error of the heliostat is small, or the heliostat is very close to the receiver plane, unacceptable errors will occur, and the peak value of the calculated flux density is four to five times the actual value. However, although the calculation results of the KGEN method can fit the actual situation to a certain extent, the calculation speed is very slow, and the calculation time is 31.3 s.

The HELIOS [8] system uses the two-dimensional fast Fourier transform method to replace convolution with function multiplication. It calculates the ellipse Gaussian

error distribution of a reflected ray generated by the ellipse Gaussian distribution of a surface error, which is suitable for more general cases. However, it consumes many computing resources. The calculated results of the HELIOS system were compared with the experimental data of a single heliostat in New Mexico [22], and the central profiles of the calculated results and the experimental data were presented. The difference between the actual data and the calculated results was approximately 1.5%. From the central profiles, the calculation error of the HELIOS system is not significant. However, the comparison of the flux density distribution involves interpolation in two directions. In the worst case, interpolation will bring about an error of approximately 2%. There is still a significant error between the calculated results of the HELIOS system and the actual data.

The numerical method proposed by the HELIOS system and Lipps has an essential assumption as a convolution method. It is necessary to assume that all reflected rays are reflected from the center of the heliostat and projected onto the receiver plane to calculate the flux density of the receiver plane. A large calculation error will occur when the distance between the receiver plane and the image plane is large. Only when the distance between the heliostat and the receiver plane is close to the focal length is the calculation result of the convolution method accurate enough. However, the direct integration method does not need to introduce this assumption in the calculation, and the error is smaller than the indirect integration. The application range of direct integration is wide. Even if the distance between the heliostat and receiver plane is unequal to the focal length, the flux density distribution can be calculated accurately.

For the convolution method we proposed before [10], we calculated the intercept factor of ten heliostats mentioned in Section 3.2. The calculated results of ten heliostats are superior to that of the UNIZAR function [3] and the HFCAL function [4]. The indirect integration's average, minimum, and maximum differences are 2.24%, 1.3%, and 2.7%. The indirect method simulates well when the intercept factor is lower than 95%, so, in most cases, it outperforms the UNIZAR and HFCAL functions.

When the intercept factor is lower than 95%, the elliptic Gaussian model proposed in this paper is consistent with the indirect integration method. When the intercept factor is higher than 95%, we can see the limitations of the convolution, and the elliptic Gaussian model performs better. As a result, the average absolute difference in the elliptic Gaussian model is 1.11% smaller than the indirect integration method. Furthermore, in Heliostat #9 and #10, where the indirect integration method performs the worst, the absolute difference is only 1.3%. This result embodies the advantage of the direct integration method. This paper has only one assumption about the sun shape, which is as close to the actual physical image as possible.

## 5. Conclusions

This paper proposes a new model for calculating the flux density distribution by a heliostat. Compared with the convolution integration method, this model uses the direct integration method with only one assumption, so it has a wide range of applicability and a high accuracy. We reconstructed the actual optical process, where reflected rays are divided into three parts, and applied Fresnel's equations to calculate the energy distribution ratios of reflected rays. The slope errors of reflected rays in the three parts are also different, which needed to be calculated separately, and were then superimposed to compute the distribution of the flux density on the receiver plane.

i    Comparing the elliptic Gaussian model with the experimental data, the results of the elliptic Gaussian model are better than SolTrace and indirect integration. The distribution of the flux density is consistent with the experimental data, and the curvature variation in the intercept factor is closer to the experimental data. The average difference in flux density distribution is 2.83%, the minimum difference is 2.33%, and the maximum difference is 3.64%; the average difference in the intercept factor is 1.30%, the minimum difference is 0.46%, and the maximum difference is 2.30%.

ii　　The circular Gaussian model simplified by the elliptic Gaussian model was compared with SolTrace. Under the condition that the reflected light intensity distribution was consistent, the flux density distribution and intercept factor obtained by the circular Gaussian model were consistent with SolTrace. The absolute differences were only 0.64% and 0.58%. However, the present model has a lower prediction ability than the SolTrace.

iii　　The integration model proposed in this paper was proven to be more accurate and applicable than convolution and function methods. The function methods are mainly based on experience and have a significant error in calculation results. Meanwhile, convolution methods themselves introduce too many assumptions, resulting in many limitations in the calculation.

iv　　This is the first time that multiple reflections and the influence of an optical error transferred from different planes of the glass mirror were considered in order to build an optical model for the flux density of a heliostat. The reflection from two surfaces of the glass mirror to form three main parts of beams was considered in the present model, and Fresnel's equations were applied to calculate the energy of the three parts of reflected rays. This may be the main reason for why more accurate results with the present model were obtained, as more reflections will add more optical errors to the reflection ray to diffuse the solar ray. As the present model has a greater prediction precision, it may be applied for the optimization of the optical field design to obtain a better design. The model applies the Gaussian model for solar brightness distribution. It is a good approximation if the optical error of the heliostat is much larger than the solar brightness distribution. However, the optical error of the heliostat gradually becomes smaller, so the model may not give a good prediction to the heliostat with a lower optical error.

**Author Contributions:** Conceptualization, Y.S.; methodology, Y.S., W.H. and L.Y.; software, Y.S.; validation, Y.S.; investigation, Y.S.; resources, Y.S.; data curation, Y.S.; writing—original draft preparation, Y.S.; writing—review and editing, B.L.; supervision, W.H. and C.Z. All authors have read and agreed to the published version of the manuscript.

**Funding:** This research received no external funding.

**Institutional Review Board Statement:** Not applicable.

**Informed Consent Statement:** Not applicable.

**Data Availability Statement:** Not applicable.

**Conflicts of Interest:** The authors declare no conflict of interest.

### List of Symbols

| | |
|---|---|
| $B$ | intensity distribution function of reflected ray |
| $d$ | the thickness of the glass |
| $E$ | the total flux density reflected by the heliostat |
| $RMES$ | the root mean square error of the flux density and intercept factor between the models and experimental data |
| $f_{int}$ | intercept factor |
| $f_0$ | focal length |
| $F$ | solar flux, $(kW/m^2)$ |
| $i_1, i_2$ | the reflection angle or refraction angle at air–glass interface |
| $i_3$ | the incident angle reaching the silver surface |
| $I$ | total intensity of reflected ray |
| $I_0$ | intensity of incidence ray |
| $I_1, I_2, I_3$ | the energy density of reflected ray of the first, second, and third part |
| $I_D$ | direct solar irradiation, $(kW/m^2)$ |
| $k$ | dielectric constant |

| $n_0$ | the ratio of refractive index of glass |
|---|---|
| $n_3$ | the ratio of refractive index of the silver plating layer relative to the glass |
| $O_1$ | grid points on the heliostat |
| $O_2$ | grid points on the receiver |
| $r$ | distance between the reflection point and the receiver point, (m) |
| $R_s$ | reflectivity of reflected ray in the S component |
| $R_p$ | reflectivity of reflected ray in the P component |
| $S_1$ | the scope of integration on the heliostat, (m$^2$) |
| $S_2$ | the scope of integration on the receiver, (m$^2$) |
| $T_s$ | transmission of reflected ray in the P component |
| $T_p$ | transmission of reflected ray in the S component |

Greek symbols:

| | |
|---|---|
| $\lambda$ | the incident angle of the solar ray to the heliostat, (rad) |
| $\beta$ | the azimuth angle of the mirror reflection point of the heliostat, (rad) |
| $\beta_1$ | the included angle between $O_1O_2$ and heliostat's normal |
| $\beta_2$ | the included angle between $O_1O_2$ and receiver's normal |
| $\alpha$ | the absorption ratio of glass to light, |
| $\theta$ | the angle variable, (mrad) |
| $\delta$ | the energy density ratios |
| $\varepsilon$ | the absorption coefficient |
| $\delta_{sun}$ | the standard deviation of the solar intensity distribution |
| $\delta_{slopex}$ | the standard deviation of the slope errors at the optical surface in transverse(x) direction, (mrad) |
| $\delta_{slopey}$ | the standard deviation of the slope errors at the optical surface in longitudinal (y) direction, (mrad) |
| $\delta_{sx}$ | the standard deviation of the optical error in transverse(x) direction |
| $\delta_{sy}$ | the standard deviation of the optical error in the Y direction |
| $\delta_x$ | the standard deviation of the optical error distribution in the X direction, (mrad) |
| $\delta_y$ | the standard deviation of the optical error distribution in the Y direction, (mrad) |
| $\bar{\delta}$ | the standard deviation of the average error |

Subscripts and superscripts:

| | |
|---|---|
| $i$ | three parts of reflected ray parameters |
| $p,s$ | the direction that is perpendicular or parallel to the vibration of the incident plane |
| *silver* | on the silver surface |
| *sun* | sun shape |
| *tr* | heliostat tracking error |

## Appendix A

*Appendix A.1. Location Determination*

In a solar tower system, the given quantities are the altitude angle $\alpha_{rn}$, the azimuth angle $\gamma_{rn}$ of the receiver plane, and the normal vector of the receiver plane $(u_{rn}, v_{rn}, w_{rn})$.

Firstly, we need to calculate the coordinates of the incident ray $(u_{sun}, v_{sun}, w_{sun})$ from the solar altitude angle $\sigma_s$ and azimuth angle $\gamma_s$:

$$(u_{sun}, v_{sun}, w_{sun}) = (cos\alpha_s sin\gamma_s, -cos\alpha_s sin\gamma_s, sin\alpha_s) \qquad (A1)$$

Then, the altitude angle $\alpha_r$ and azimuth angle $\gamma_r$ of the central reflected ray can be calculated by the position of the heliostat and the receiver plane:

$$\alpha_r = atan(\frac{Z_{receiver} - Z_{heliostat}}{\sqrt{(x_{receiver} - x_{heliostat})^2 + (y_{receiver} - y_{heliostat})^2}}) \qquad (A2)$$

$$\gamma_r = atan(\frac{x_{receiver} - x_{heliostat}}{y_{receiver} - y_{heliostat}}) \qquad (A3)$$

where $(x_{heliostat}, y_{heliostat}, z_{heliostat})$ and $(x_{receiver}, y_{receiver}, z_{receiver})$ represent the heliostat center coordinates and receiver center coordinates in the global coordinate system, respectively.

The altitude angle $\alpha_n$ and azimuth angle $\gamma_n$ of the heliostat normal vector should also be calculated:

$$\alpha_n = atan\left(\frac{sin\alpha_s + sin\alpha_r}{\sqrt{cos^2\alpha_s + cos^2\alpha_r + 2cos\alpha_s cos\alpha_r cos(\gamma_r - \gamma_s)}}\right) \tag{A4}$$

$$\gamma_n = atan\left(\frac{cos\alpha_r sin\gamma_r + cos\alpha_s sin\gamma_s}{cos\alpha_r cos\gamma_r + cos\alpha_s sin\gamma_s}\right) \tag{A5}$$

*Appendix A.2. Meshing Process and Coordinate System Rotation*

The next step is to grid the heliostat and the receiver plane into a 100-by-100 grid, and the center point of each grid, namely the grid point, is used to represent the grid. The unit grid of a heliostat is expressed by Δhg, and the unit grid of a heliostat is expressed by Δrg. Then, we calculated the coordinates of the heliostat and the receiver plane in the coordinate system.

In the receiver plane coordinate system, the coordinate of the receiver plane is $(x_r, y_r, z_r)_{receiver}$, where $x_r$ and $y_r$ were obtained by meshing the length and width of the receiver plane. Since the receiver plane is flat, the $z_r$ coordinate is always 0.

In the coordinate system of the heliostat, the coordinate of the heliostat is $(x_r, y_r, z_r)_{heliostat}$, where $x_h$ and $y_h$ were obtained by meshing the length and width of the heliostat. Since the heliostat is spherical, the $z_h$ coordinate is:

$$z_h = \sqrt{4f_0^2 - x_h^2 - y_h^2} \tag{A6}$$

After completing the meshwork of the heliostat and receiver plane, the next step is to transform the heliostat coordinate system and receiver plane coordinate system into the global coordinate system. The total rotation matrix from the global coordinate system to the heliostat coordinate system is:

$$AH = \begin{bmatrix} cos\gamma_n & -sin\gamma_n & 0 \\ sin\gamma_n sin\alpha_n & cos\gamma_n sin\alpha_n & cos\alpha_n \\ sin\gamma_n cos\alpha_n & -cos\gamma_n cos\alpha_n & sin\alpha_n \end{bmatrix} \tag{A7}$$

Thus, the coordinate of the heliostat grid point in the global coordinate system is:

$$(x_h, y_h, z_h)_{global} = (x_{heliostat}, x_{heliostat}, x_{heliostat})_{global} + (x_h, y_h, z_h)_{heliostat} \times AH^{-1} \tag{A8}$$

Similarly, the rotation matrix from the global coordinate system to the receiver plane coordinate system is:

$$AR = \begin{bmatrix} cos\gamma_{rn} & -sin\gamma_{rn} & 0 \\ sin\gamma_{rn} sin\alpha_{rn} & cos\gamma_{rn} sin\alpha_{rn} & cos\alpha_{rn} \\ sin\gamma_{rn} cos\alpha_{rn} & -cos\gamma_{rn} cos\alpha_{rn} & sin\alpha_{rn} \end{bmatrix} \tag{A9}$$

After gridding the heliostat, we can further determine the coordinates of the heliostat spherical center:

$$x_{spherical} = x_{heliostat} + 2f_0(-cos\alpha_n sin\gamma_n) \tag{A10}$$

$$y_{spherical} = y_{heliostat} + 2f_0(-cos\alpha_n cos\gamma_n) \tag{A11}$$

$$z_{spherical} = z_{heliostat} + 2f_0 sin\alpha_n \tag{A12}$$

Further determining the normal vector of each point on the heliostat:

$$u_{hn} = \frac{x_{spherical} - x_h}{l_{so}} \tag{A13}$$

$$v_{hn} = \frac{y_{spherical} - y_h}{l_{so}} \tag{A14}$$

$$w_{hn} = \frac{z_{spherical} - z_h}{l_{so}} \tag{A15}$$

where $l_{so}$ is the distance between the heliostat grid point and the heliostat spherical center, calculated as follows:

$$l_{so} = sqrt(x_{spherical} - x_h)^2 + (y_{spherical - y_h})^2 + (z_{spherical} - z_h)^2 \tag{A16}$$

*Appendix A.3. Calculation of Geometrical Optics*

Then, the reflected ray can be calculated according to the incident vector $(u_{sun}, v_{sun}, w_{sun})$ and the normal vector of the heliostat grid point $(u_{hn}, v_{hn}, w_{hn})$:

$$u_{ray} = 2(u_{hn}u_{sun} + v_{hn}v_{sun} + w_{hn}w_{sun})u_{hn} - u_{sun} \tag{A17}$$

$$v_{ray} = 2(u_{hn}u_{sun} + v_{hn}v_{sun} + w_{hn}w_{sun})v_{hn} - v_{sun} \tag{A18}$$

$$w_{ray} = 2(u_{hn}u_{sun} + v_{hn}v_{sun} + w_{hn}w_{sun})w_{hn} - w_{sun} \tag{A19}$$

The receiver point, which coordinates in the global coordinate system, is formed when a reflected ray hits the receiver plane:

$$t = \frac{z_{receiver}w_{rn} + y_{receiver}v_{rn} + x_{receiver}u_{rn} - u_{rn}x_h - v_{rn}y_h - w_{rn}z_h}{u_{rn}u_{ray} + v_{rn}v_{ray} + w_{rn}w_{ray}}, \tag{A20}$$

$$x_{tf} = x_h + u_{ray} \times t \tag{A21}$$

$$y_{tf} = y_h + v_{ray} \times t \tag{A22}$$

$$z_{tf} = z_h + w_{ray} \times t \tag{A23}$$

The receiver point $O_3$ on the receiver is converted from the global coordinate system to the receiver plane coordinate system:

$$(x_{rf}, y_{rf}, z_{rf})_{receiver} = (x_{rf} - x_{receiver}, y_{rf} - y_{receiver}, z_{rf} - z_{receiver})_{global} \times AR. \tag{A24}$$

Heliostat grid point coordinates are converted from the global coordinate system to the receiver plane coordinates:

$$(x_h, y_h, z_h)_{receiver} = (x_h - x_{receiver}, y_h - y_{receiver}, z_h - z_{receiver})_{global} \times AR \tag{A25}$$

The $O_1O_2$ vector is:

$$u_{O_1O_2} = -(x_r - x_h) \tag{A26}$$

$$v_{O_1O_2} = -(y_r - y_h) \tag{A27}$$

$$w_{O_1O_2} = -(z_r - z_h) \tag{A28}$$

The angle between $O_1O_2$ and the heliostat normal vector is:

$$\beta_1 = acos(\frac{u_{O_1O_2}u_{hn} + v_{O_1O_2}v_{hn} + w_{O_1O_2}w_{hn}}{\sqrt{u_{O_1O_2}^2 + v_{O_1O_2}^2 + w_{O_1O_2}^2}}) \tag{A29}$$

The angle between $O_1O_2$ and the receiver normal vector is:

$$\beta_2 = acos(\frac{u_{O_1O_2}u_{rn} + v_{O_1O_2}v_{rn} + w_{O_1O_2}w_{rn}}{\sqrt{u_{O_1O_2}^2 + v_{O_1O_2}^2 + w_{O_1O_2}^2}}) \tag{A30}$$

Firstly, we calculated the distances between points in the coordinate system of the receiver plane:

(1)  The distance between receiver point $O_3$ and heliostat grid point $O_1$ is:

$$l_{hrf} = \sqrt{(x_h - x_{rf})^2 + (y_h - y_{rf})^2 + (z_h - z_{rf})^2} \tag{A31}$$

and $l_{hrf}$ in the X direction and the Y direction are, respectively:

$$l_{hrfx} = \sqrt{(x_h - x_{rf})^2 + (z_h - z_{rf})^2} \tag{A32}$$

$$l_{hrfy} = \sqrt{(y_h - y_{rf})^2 + (z_h - z_{rf})^2} \tag{A33}$$

(2)  The distance between receiver point $O_3$ and receiver plane grid point $O_2$ is:

$$l_{rfr} = \sqrt{(x_r - x_{rf})^2 + (y_r - y_{rf})^2 + (z_r - z_{rf})^2} \tag{A34}$$

and $l_{rfr}$ in the X direction and the Y direction are, respectively:

$$l_{rfrx} = \sqrt{(x_r - x_{rf})^2 + (z_r - z_{rf})^2} \tag{A35}$$

$$l_{rfry} = \sqrt{(y_r - y_{rf})^2 + (z_r - z_{rf})^2} \tag{A36}$$

(3)  The distance between heliostat grid point $O_1$ and receiver plane grid point $O_2$ is:

$$l_{rh} = \sqrt{(x_r - x_h)^2 + (y_r - y_h)^2 + (z_r - z_h)^2} \tag{A37}$$

and $l_{rh}$ in the X direction and the Y direction are, respectively:

$$l_{rhx} = \sqrt{(x_r - x_h)^2 + (z_r - z_h)^2} \tag{A38}$$

$$l_{rhy} = \sqrt{(y_r - y_h)^2 + (z_r - z_h)^2} \tag{A39}$$

Then, the included angle between the reflected ray and $O_1O_2$ of each grid can be calculated, as shown in Figures A1–A3:

$$\theta_x = acos(\frac{l_{rhx}^2 + l_{hrfx}^2 - l_{rfrx}^2}{2l_{rhx}l_{hrfx}}) \tag{A40}$$

$$\theta_y = acos(\frac{l_{rhy}^2 + l_{hrfy}^2 - l_{rfry}^2}{2l_{rhy}l_{hrfy}}) \tag{A41}$$

$$\theta = acos(\frac{l_{rh}^2 + l_{hrf}^2 - l_{rfr}^2}{2l_{rh}l_{hrf}}) \tag{A42}$$

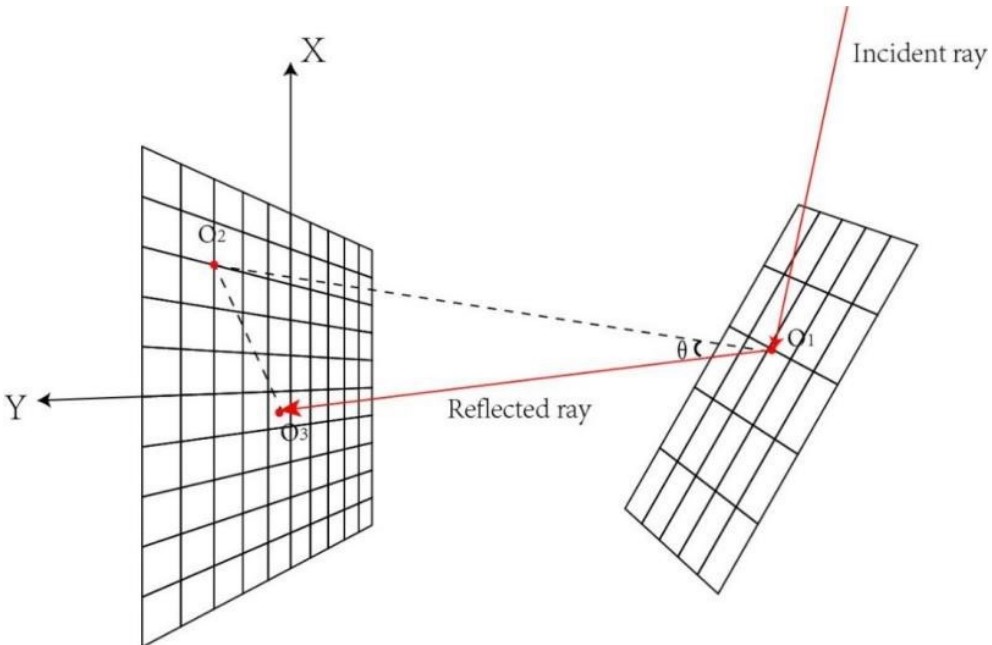

**Figure A1.** The included angle between the reflected ray and $O_1O_2$ of each grid under the circular Gaussian model.

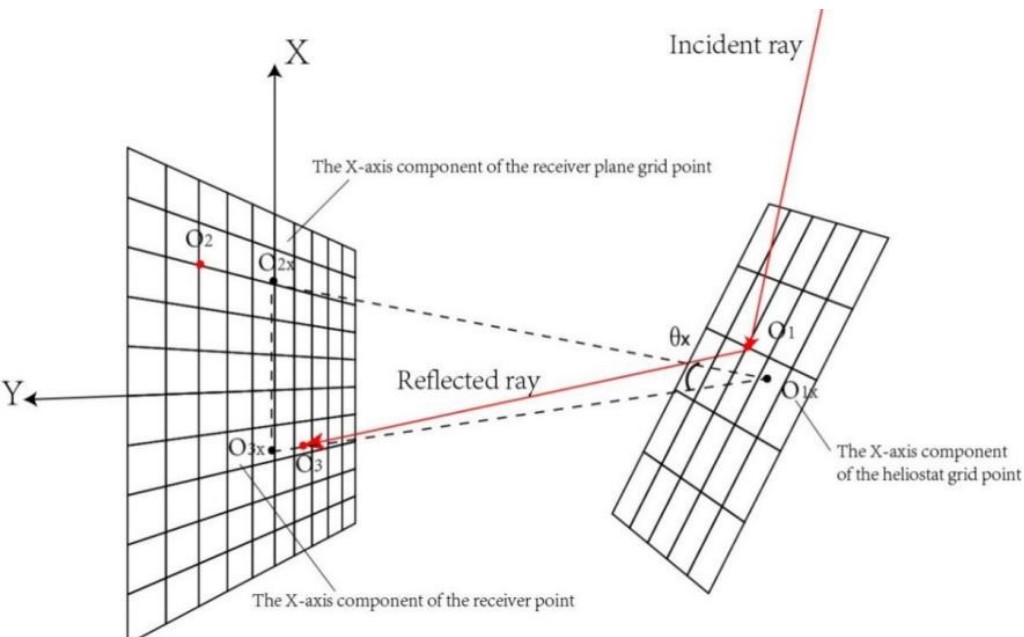

**Figure A2.** The included angle between the reflected ray of each grid and $O_1O_2$ in the X-direction under the elliptic Gaussian model.

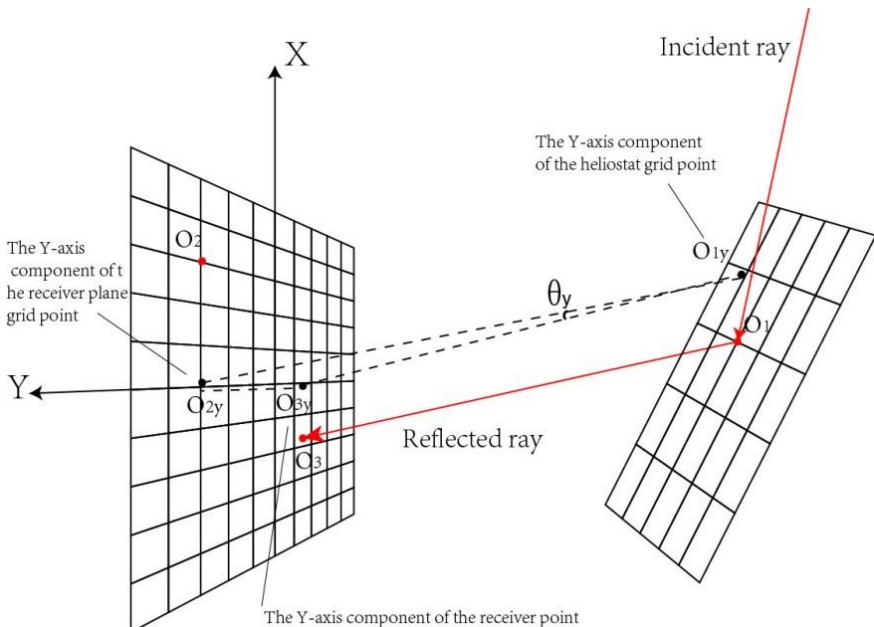

**Figure A3.** The included angle between the reflected ray of each grid and $O_1O_2$ in the Y-direction under the elliptic Gaussian model.

*Appendix A.4. Solution of the Flux Density at a Specific Point*

Finally, considering the sun shape and optical errors, the distribution function of reflected rays on the plane of the receiver was computed:

Circular Gaussian model:

$$B(\theta) = \frac{1}{2\pi\sigma^2} \exp(-\frac{\theta^2}{2\sigma^2}) \tag{A43}$$

Elliptic Gaussian model:

$$B(\theta_x, \theta_y) = \frac{1}{2\pi\sigma_x\sigma_y} \exp[-\frac{1}{2}(\frac{\theta_x^2}{\sigma_x^2} + \frac{\theta_y^2}{\sigma_y^2})] \tag{A44}$$

Therefore, the contribution of the heliostat to the flux density of the grid point on the receiver plane is:

Circular Gaussian model:

$$F_{u,v} = \frac{I\cos\beta_1\cos\beta_2 B(\theta) \times \Delta hg}{r^2} \tag{A45}$$

Elliptic Gaussian model:

$$F_{u,v} = \frac{I\cos\beta_1\cos\beta_2 B(\theta_x, \theta_y) \times \Delta hg}{r^2} \tag{A46}$$

where $F_{u,v}$ is the contribution of grid points in row $u$ and column $v$ on the heliostat to the flux density value of a point on the receiver plane. Finally, the flux density value of a point on the receiver plane can be obtained by integrating all grid points on the heliostat.

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
