# Peer review of "An Integration Model for Flux Density Distribution Formed by a Heliostat"

_applsci, doi:10.3390/app122010191_

Round 1

Reviewer 1 Report

What practical importance does this work have?

P.1 – Why are you using Gauss model for flux density distribution?

Pp.1-2: Introduction: OK;

P.3, R: 81-89: Put references for those values for I1 (5%), I2 (91.5%) and I3 (3%);

P.8, R. 205: Why compare the results with Solar Trace?

P.8, Eq.(24): Why slope error is calculated in this form?

P.9, Eq.(28): Put references;

P.10, Table 1: That are parameters from a real heliostat?

P.11, Fig.6: Explain better this figure;

P.11, R. 283-284: Where do these values ​​for I1, I2 and I3 come from?

P.12, Figs.7-16: The circular Gaussian model is in the left side and SolTrace model is in the right side?

P.16, Section 4. Analysis and discussion is OK;

P.18, Section 5. Conclusion: What will be the next research based on the results of this work?

Reviewer 2 Report

Review of: An integration model for flux density distribution formed by a heliostat.

Manuscript number: applsci-1865098

Overall recommendation: Major Revision

Overview and general recommendation:

An integration model to calculate the flux density distribution after only applying the Gauss model for solar brightness distribution. It is the first time to consider the multiple reflections and the influence of optical error transferred from different planes of the glass mirror to build an optical model for flux density of heliostat. The average relative prediction error of the present model from the experimental data is only 2.83%, less than SolTrace and other models.

Major comments:

1-  Manuscript English language editing is required. There are numerous cases of minor grammatical inaccuracies and typos.

2- Please rewrite the abstract by identifying the purpose, the problem, the methodology and the important results (not all) and conclusions of your work.

3-  The originality and the novelty of this paper are not clear, and they all should be further justified and improved. 

4- Please update the survey of related work literature. The updated literature review must put the problems of this work into a meaningful context and contain the latest insights on research in the field.

5- In the literature review, the author must put some results for the reference cited.

6- The authors can consider this paper in your work (doi.org/10.3390/su12135392)

7-      In all Equations, please delete the comma and dot at the end of Eqs.

8-      Why is this method and model an effective way to overcome the limitations and operational problems?

9-      Try to summaries the section of Analysis and discussion in table or in Figure for comparison and to simplicity shows the results of different techniques.  

10-  The conclusion section needs more clarity.  

11-  Add some recommendation statements at the end of the conclusion part.

12-  Start table of symbol by (List of Symbol).

13-  Revise and follow the guide to authors. No one of the references is quoted correctly. It is advisable to add the DOI.

I hope these comments will be helpful to you.

Reviewer 3 Report

Review Report on Manuscript ID: applsci-1865098

Recommendation: minor revision

Type: Article

Title:  An integration model for flux density distribution formed by a heliostat

Authors:   Chenggang Zong,Yemao Shi, Liang Yu, Bowen Liu

Corresponding Auhor:  Weidong Huang

 General statement

It is an intersting and valuable paper. The Authors’ proposed an oryginal  integral model to calculate the flux density distribution after only applying the Gauss model for solar brightness distribution. An element of novelty is consideration of the multiple reflections and the influence of optical error transferred from different planes of the glass mirror to build an optical model for flux density of heliostat. Experimental validation of the elliptic Gaussian model showed a maximum relative error of 3.64%. In my opinion the present form of the manuscript needs  only minor revision which results from the comments below.

 Remarks

1      Line 108, I suggest replacing the citation [11] with M. Born and E. Wolf, Principles of Optics, 60th anniversary edition 2019. Cambridge University Press (p.44, Eq. 30)

2     Line 119, From [12] Table I,  we have for Silver: for n=0.03 , k=5.242, for n=0.04 k=4.838. From which follows the value of k=3.9046 for n=0.03416 ?

3      Line 165, Please provide page number for citation [7].

4      Line 259, Figure 5 (b), I suggest enriching this graph with residues

5      Line 289, Editorial error, instead of Fig.s 7 to 16 it should be Figs. 7 to 16

6      Line 363, instead of 3.6 GHZ it should be 3.6 GHz

Round 2

Reviewer 2 Report

Dear Author, Thank you for revised Paper.

Pleas complete these minor comments

1- The authors can consider this paper in your work for complete incident radiation (doi.org/10.3390/su12135392).

2- In all Equations, please delete the comma and dot at the end of Eqs. (also, take Appendix in consideration).

3- 
